# RACCOON: REGRET-BASED ADAPTIVE CURRICULA FOR COOPERATION

## ABSTRACT

Overfitting to training partners is a common problem in cooperative multi-agent reinforcement learning, leading to poor zero-shot transfer to novel partners. A popular solution is to train an agent with a *diverse* population of partners. However, previous work lacks a principled approach for selecting partners from this population during training, usually sampling at random. We argue that partner sampling is an important and overlooked problem, and motivated by the success of regret-based Unsupervised Environment Design, we propose *Regret-based Adaptive Curricula for **Coo**peration* (RACCOON), a novel method which prioritises high-regret partners and tasks. We test RACCOON in the Overcooked environment, and demonstrate that it leads to sample efficiency gains and increased robustness across diverse partners and tasks, compared with strong baselines. We further analyse the nature of the induced curricula, and conclude with discussions on the limitations of cooperative regret and directions for future work.

## 1 INTRODUCTION

Many real-world problems require collaboration between two or more agents to achieve a common goal, where agents may be autonomous machine learning agents or humans (Dafoe et al., 2021). A popular approach to training such agents is cooperative multi-agent reinforcement learning (MARL), in which multiple agents interact and learn to maximise a common reward (Albrecht et al., 2024; Oroojlooy & Hajinezhad, 2023; Papoudakis et al., 2020). However, such agents also need to be able to adapt to partners with diverse preferences and abilities (Siu et al., 2021). This is the problem of *ad-hoc teamwork*: developing agents which can efficiently and robustly succeed with unseen partners (Stone et al., 2010; Mirsky et al., 2022).

While *self-play* (SP, Tesauro et al. (1995)) has demonstrated its effectiveness for training agents with strong transfer to unseen opponents in two-player zero-sum settings such as Go (Silver et al., 2017), Diplomacy (Bakhtin et al., 2022) and poker (Brown & Sandholm, 2019), SP alone fails to produce robust team players in fully cooperative settings Carroll et al. (2019); Charakorn et al. (2020). Common-reward games, unlike two-player zero-sum games, can admit a number of incompatible equilibria, and SP policies trained on collaborative tasks tend to rely on efficient but arbitrary conventions which render them poor teammates when paired with novel partners (Lowe et al., 2019; Hu et al., 2020).

A popular alternative is to train an agent—which we refer to as the *student*—with *diverse pre-trained partners* for a given task, under the assumption that exposure to diverse partner behaviours during training leads to better generalisation. As a result, much work on ad-hoc teamwork is concerned with obtaining a diverse, high-quality set of training partners. A central open challenge is generating *meaningful* diversity—ensuring that behaviours don't differ in merely superficial ways—while maintaining *reasonable* behaviour. Previous approaches include relying on diversity introduced by varying initial seeds and architectures Strouse et al. (2021), maximising statistical divergence between actions or trajectories (Lupu et al., 2021; Zhao et al., 2023) or minimising *cross-play* (XP) performance between partners in the population (Charakorn et al., 2022; Cui et al., 2022; Sarkar et al., 2024; Rahman et al., 2022). However, while generating partner diversity is an important problem, this is not the problem we address in this paper. Instead we ask: **given a population of partners, *how can we best use this population to promote the student's learning?*** Even in a diverse, reasonable partner pool, it is extremely unlikely that all partners are equally useful for learning throughout

training—in other words, that their *learning potential* is uniform across the population and static across time—as the student improves. However, previous work implicitly makes this assumption by sampling partners uniformly at random during the student's training (Strouse et al., 2021; Sarkar et al., 2024; Charakorn et al., 2022). In addition, current methods for generating diverse partner populations are likely to be imperfect, for example including duplicates of certain behaviours or uncooperative partners, since it is still unclear what constitutes relevant diversity for a given problem. This further implies that all partners should blindly not be assigned equal weight. We argue that deciding how to prioritise partners requires further careful consideration.

Inspired by recent successes of regret-based Unsupervised Environment Design (UED) in single-agent RL and two-player zero-sum MARL (Dennis et al., 2020; Jiang et al., 2021; Samvelyan et al., 2023), we propose an autocurriculum that prioritises *high-regret partners*, where a partner's regret at a given time is defined as the difference between the optimal and current XP return with that partner. Our novel replay-based method, RACCOON (**R**egret-based **A**daptive **C**urricula for **Coo**peratio**n**), uses a *relative regret* metric to estimate the learning potential of partners and tasks in a way which reflects their relative difficulty. We empirically demonstrate the robustness and sample efficiency gains of RACCOON—compared with baselines which randomise over partners or maximise worst-case performance—in the two-player fully cooperative Overcooked environment, a popular cooperative benchmark in which players must collaborate to cook and deliver soup (Carroll et al., 2019).[1] We additionally analyse the induced curricula and perform an ablation which verifies the importance of using *relative* rather than absolute regret for improved ad-hoc teamwork.

As RACCOON can be paired with any partner population and set of tasks, our method complements work on diverse partner generation. We hope that our work motivates the problem of finding effective curricula over partners for cooperation, and provides a starting point for further research on cooperative autocurricula.

## 2 BACKGROUND

**Cooperative MARL**   We consider the *fully cooperative* setting in which a number of agents interact in an environment and receive common rewards. Since we also allow for varying tasks or "levels" in the environment, following the UED literature we model this as an $n$-agent *decentralised underspecified partially observable Markov decision process* (Dec-UPOMDP), described as a tuple $\langle \mathcal{S}, \mathcal{A}, \mathcal{T}, R, \gamma, T, \Theta \rangle$, where $\mathcal{S}$ is the state space, $\mathcal{A} = \{A_i\}_{1 \leq i \leq n}$ is the joint action space, $\mathcal{T} : \mathcal{S} \times \mathcal{A} \times \mathcal{S} \to [0, 1]$ is the transition function with reward function $R : \mathcal{S} \times \mathcal{A} \to \mathbb{R}$, $\gamma$ is the reward discount factor, T the horizon and $\Theta$ the set of free parameters of the environment. In this case, the transition function $\mathcal{T}$ additionally takes an environment configuration $\theta \in \Theta$ as an argument.

**Ad-hoc teamwork**   The aim of ad-hoc teamwork, introduced by Stone et al. (2010), is to achieve collaboration without prior coordination. Two common approaches are *modelling other agents* and *training with diverse partners* (Mirsky et al., 2022). We consider the latter approach in this work, in which an agent, which we refer to as the **student**, is trained with a population of pre-trained agents, which we refer to as the **partners** (Charakorn et al., 2020; Lupu et al., 2021; Cui et al., 2022; Rahman et al., 2022; Sarkar et al., 2024). These partners are typically trained via SP with some form of diversity regularisation. A related problem is *zero-shot coordination*, which additionally assumes that agents are trained independently using the same algorithm, and thus measures a particular algorithm's ability to break symmetries in non-arbitrary ways (Hu et al., 2020).

**Unsupervised Environment Design**   Given an environment with configurable parameters—each defining a separate *level* (i.e., task)—UED frames curriculum generation as a game between a **teacher** and a **student** where the teacher iteratively chooses levels for the student to train on (Dennis et al., 2020). In regret-based UED, the teacher's objective is to maximise *regret* of the student—how much worse the student performs as compared with an optimal policy—while the student aims to maximise return as usual. This has been shown to induce a curriculum which presents the student with the simplest tasks it cannot yet solve (Dennis et al., 2020). *Prioritised Level Replay* (PLR), the prevailing UED method, curates a curriculum for the student by storing and replaying the highest-regret levels drawn from a random level generator (Jiang et al., 2021).

---

[1]Code can be found at https://anonymous.4open.science/r/raccoon.

# 3 METHOD

## 3.1 COOPERATIVE REGRET AS LEARNING POTENTIAL

Given a fixed population of partners and a task space, we desire two properties for an effective curriculum: it should adapt to the student's changing ability throughout training, and it should reflect each (partner, task) pair's relative *learning potential* for the student—a term we use informally to denote how much progress the student can make towards improved ad-hoc teamwork by training with that partner on that task.

Randomising uniformly over partners lacks both these properties; while successful (Strouse et al., 2021; Sarkar et al., 2024), it likely leaves sample efficiency gains on the table as certain partners become easy to cooperate with and are unnecessarily resampled. Following UED literature, we refer to this approach as *domain randomisation* (DR), where here the domain is the partner pool.

An alternative is to prioritise partners with whom the student currently has *lowest* XP return. This is employed in Zhao et al. (2023) and partly addresses the intuition that partners who pose a more difficult coordination problem should be sampled more frequently. In the UED framing, this teacher is a *minimax adversary*. However, in task-based UED, the minimax adversary prioritises levels which are impossible to solve, and therefore confer no learning benefit. Similarly, in the current setting, a minimax adversary will prioritise partners with whom it's impossible to cooperate. In addition, the minimax adversary does not reflect differences in maximum achievable returns with each partner. If there are partners with whom it's impossible to succeed, then the minimax adversary will continue to prioritise those partners at the expense of the student training with other, more useful partners.

We propose a third sampling method: **maximising regret**. For a given decision problem, the *regret* of a policy is defined as the difference between the best possible outcome which could have been obtained in that problem, and the actual outcome obtained by the policy (Savage, 1951). For a fixed task in the two-player multi-agent setting, we define the regret of a policy $\pi$ with partner $\pi'$ as

$$Regret(\pi, \pi') = U(BR(\pi'), \pi') - U(\pi, \pi')$$

where $U(\pi, \pi')$ is the expected return obtained by $(\pi, \pi')$, and $BR(\pi')$ denotes a best response to $\pi'$.

A curriculum which prioritises high-regret partners has both our desired properties: it adapts to the student's ability, and it and provides a reasonable measure of *learning potential* by measuring the gap between current performance and optimal performance with each partner. Furthermore, at equilibrium in the teacher-student game, regret-based UED provably results in the student implementing a minimax regret policy, which performs near-optimally with everyone if it is possible to do so (Dennis et al., 2020). Arguably, for ad-hoc teamwork, we want to find a policy which succeeds with as many partners as possible, even if it doesn't get the highest return with each individual one.

It's worth nothing that the effectiveness of regret requires that there is a single policy which can succeed with all partners with whom success is possible (Dennis et al., 2020), which may not exist in settings where partners display incompatible conventions such that it is impossible to cooperate with everyone under uncertainty about the partner's policy. We discuss limitations of regret in Section 6.

## 3.2 RACCOON

Motivated by the above discussion, we propose RACCOON (**R**egret-based **A**daptive **C**urricula for **Coo**peratio**n**), a replay-based method which considers distinct tasks as well as partners for maximal generality. Following Samvelyan et al. (2023), RACCOON maintains a joint buffer over partners $\Pi$ and tasks $\Theta$, with the key difference being that we use *pre-trained policies* as partners, rather than past checkpoints of the student policy. The buffer stores tasks and associated *scores $S$* for each partner, where as scores we use regret estimates (see more in Section 3.3). At the beginning of each episode, the scores are used to obtain a distribution over partners, $\Delta_S(\Pi)$, from which a partner is sampled. We use a rank-based prioritisation, in which the partners' mean scores are ranked, and the probability of sampling a partner $\pi_i$ is proportional to the inverse of its rank $n_i$, adjusted by temperature $\beta$:

$$P(\pi_i | S) \propto \frac{(1/n_i)^{1/\beta}}{\sum (1/n_i)^{1/\beta}}.$$

A task is either replayed from the sampled partner $\pi_i$'s buffer $\Lambda_i$ or randomly generated, controlled by a *replay probability* hyperparameter. If replaying from $\Lambda_i$, the scores $S_i$ in the buffer are ranked, and

the inverse ranks similarly provide a distribution over the tasks in the buffer, $\Delta_{S_i}(\Lambda_i)$, from which the replayed task is sampled. New tasks are added to the buffer if their scores are sufficiently high, and scores for replayed tasks are updated until those environments are replaced by new higher-scoring tasks. RACCOON is agnostic to the pool of partner agents, instead seeking to leverage that pool to train a maximally robust student agent.

Pseudocode is provided in Algorithm 1. $\Delta_U$ refers to the uniform distribution.

---

**Algorithm 1** RACCOON

---

**Input:** Pre-trained partners $\Pi$, environment parameters $\Theta$
Initialise student policy $\pi$ and empty buffer $\Lambda$ with scores $S = 0$
**while** *not converged* **do**
$\quad \pi_i \sim \Delta_S(\Pi)$                  // Sample partner according to scores
$\quad$ Sample replay decision
$\quad$ **if** *replaying* **then**
$\quad\quad \theta \sim \Delta_{S_i}(\Lambda_i)$            // Sample task from partner buffer
$\quad$ **else**
$\quad\quad \theta \sim \Delta_U(\Theta)$             // Sample a uniform random task
$\quad$ Collect trajectory $\tau$ on $\theta$ using $(\pi, \pi_i)$
$\quad$ Compute regret score $S = Regret^\theta(\pi, \pi_i)$
$\quad$ Update $\pi$ with rewards $R(\tau)$
$\quad$ (Optionally) Update $\Lambda_i$ with $\theta$ using score $S$

---

### 3.3 ESTIMATING PARTNER REGRET

In practice, we typically do not have access to a best response to each partner policy and instead need to estimate the highest achievable return with each partner in order to estimate regret. We discuss using a learned best response and other methods in Appendix A, but we find the most effective method estimates regret of partner $\pi'$ (on a task) as the difference between the maximum return ever achieved with $\pi'$ (on that task) and the current return. In order to reflect different achievable returns for different partners and tasks, we normalise to obtain **relative regret**, defined as

$$Score^\theta(\pi, \pi_i) = \frac{R^\theta_{\max}(\pi, \pi_i) - \sum_{t=0}^{T} r_t}{R^\theta_{\max}(\pi, \pi_i)}$$

for student $\pi$ and partner $\pi_i$, where $r_t$ are the rewards from the most recent trajectory of $(\pi, \pi_i)$ on $\theta$, and $R^\theta_{\max}(\pi, \pi_i)$ is the maximum return previously achieved by $(\pi, \pi_i)$ on $\theta$. This scoring function requires no prior knowledge about the partners' abilities and effectively "bootstraps" the estimate of optimal return with each partner as the student learns, getting more accurate as the student policy improves. Using the maximum return ever achieved with each partner also prevents the student from later forgetting how to cooperate with a partner.

## 4 EXPERIMENTAL SET-UP

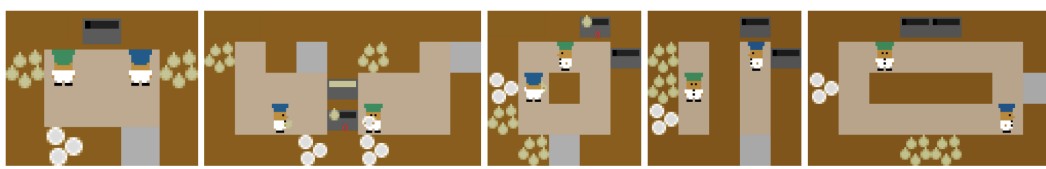

Figure 1: **Overcooked layouts.** Left to right: Cramped Room, Asymmetric Advantages, Coordination Ring, Forced Coordination, Counter Circuit.

**Environment.** We test RACCOON in Overcooked (Carroll et al., 2019), a collaborative, fully-observable two-player game in which players work together to cook and deliver onion soup. Over-cooked admits many different playing conventions (Sarkar et al., 2024), such as how tasks are divided

between players, making it non-trivial to learn a policy which can adapt to arbitrary unseen partners. In addition, although Overcooked is a two-player game, the two-player setting is interesting for studying coordination because each player has more potential to shape the behaviour of the other player and influence equilibrium selection. Episodes last a fixed number of steps and players receive a common reward for each soup delivery. Overcooked has five standard layouts, shown in Figure 1, each presenting distinct challenges such as avoiding collisions or uneven division of labour. Initial positions of players and objects are fixed, but we vary whether the student controls Player 1 or Player 2 (blue or green hat) since different roles can require different behaviours. We base our environment on the implementation of Overcooked-AI from the JaxMARL library (Rutherford et al., 2024) and implement RACCOON by building on Prioritised Level Replay from the Minimax library for UED (Jiang et al., 2023).

**Baselines and ablations.** We compare RACCOON—using the score described in Section 3.3—with two strong baselines: domain randomisation (DR, Jakobi (1997)), which uniformly randomises over partners and tasks, and minimax adversary (Minimax, Pinto et al. (2017); Morimoto & Doya (2005)), where we replace the regret estimate with the negative return as the score in RACCOON. All students are trained and evaluated with the same sets of training and test partners. We also investigate the importance of using *relative* regret by performing an ablation, which we denote as RACCOON⁻, in which we use *absolute* (unnormalised) regret as the score.

**Generating partners.** Following Strouse et al. (2021), we obtain diverse partners by training policies in SP with different initial seeds, and using checkpoints from the beginning, middle and end of training to cover a range of skill levels, reflecting the fact that novel partners such as humans are unlikely to behave optimally. For each layout $L$, we train a population of partners $\Pi_L$ on $L$. We choose to train partners in this "specialist" way, rather than training on all layouts simultaneously, as we find it results in higher quality policies for each layout, and further allows for faithful selection of partners of the three skill levels.

We form the training partner population $\Pi_L^{\text{train}}$ by taking five seeds from $\Pi_L$ and checkpoints from the beginning of training, at convergence and when returns are half the final return, resulting in "low-", "medium-" and "high-skilled" partners, following the convention in Strouse et al. (2021). Note that this means that the low-skilled training partners follow random policies. Similarly, we form the held-out partner population $\Pi_L^{\text{test}}$ using five different seeds and three checkpoints, but we use slightly later checkpoints than for $\Pi_L^{\text{train}}$ to model the assumption that unseen partners in ad-hoc teamwork should be at least better than random at the task. We use PPO and an actor-critic network with a shared convolutional layer for both student and partner policies; for more details see Appendix C.

Despite the existence of more complicated methods for generating diversity, varying initial seeds has been shown to be effective despite its simplicity (Charakorn et al., 2020), and we expect the range of skill levels afforded by different checkpoints to pose a particularly interesting setting in which to investigate curricula, due to the strong asymmetries in learning potential between partners.

**Tasks.** We test RACCOON on both *multi-task* and *single-task* settings. For multi-task settings, we use all five Overcooked layouts and both player positions, resulting in a task pool of size 10. These are padded to the same size so that the student can train on all tasks simultaneously. We primarily focus on the multi-task setting, because we are interested in RACCOON's ability to find effective learning opportunities from many possible combinations of partenrs and tasks. We use the single-task setting to analyse the induced curriculum over partners only, and single-task results can be found in Appendix B.2.

# 5 RESULTS

**Robustness across multiple tasks.** In the multi-task setting, the student trains on all five layouts in both player positions, and we use a training partner population of size 75 by aggregating $\Pi_L^{\text{train}}$ for all $L$. Note that since partners are trained in SP, they can play either position for their layout. At the start of each episode, a partner, layout and student player position is sampled, and we train each student for 400M environment steps.

Table 1: **Returns with held-out partners for students trained on all five layouts simultaneously.** For each episode, one of 75 training partners, a layout and player roles are sampled. Students are then evaluated separately on each layout with 15 diverse held-out partners, who have been trained on that layout to varying skill levels. We calculate the mean return across partners over 20 trials per layout (10 per player position), and report the mean and standard error for five independent training runs for each method. RACCOON⁻ is an ablation of RACCOON using absolute rather than relative regret. **Bold** values are within one standard error of the best mean.

| Method | Cramped Rm. | Asymm. Adv. | Coord. Ring | Forced Coord. | Counter Circ. |
|---|---|---|---|---|---|
| DR | **184.4 ± 0.8** | **208.2 ± 6.6** | 27.3 ± 3.3 | 0.6 ±0.21 | 0.6 ±0.1 |
| Minimax | 180.4 ±1.0 | 199.7 ±14.0 | **42.9 ± 11.85** | 4.6 ±2.93 | 4.4 ±2.1 |
| RACCOON | 166.3 ±2.5 | **209.7 ± 3.8** | **43.5 ± 9.4** | **35.0 ± 9.1** | **22.2 ± 3.2** |
| RACCOON⁻ | 177.2 ± 1.1 | 204.5 ± 4.1 | **43.1 ± 6.5** | 0.3 ± 0.1 | 0.7 ± 0.1 |

Figure 2 shows the average return of the student with $\Pi_L^{\text{test}}$ for each layout $L$ in each position. RACCOON performs competitively with DR and Minimax on the easier first three layouts, and notably is able to make at least one delivery on average in Forced Coordination and Counter Circuit, where DR and Minimax consistently fail to make any at all. Because the training partner pool consists of "specialist" partners trained on individual layouts to a degree of skill levels, the learning potential of partner-layout pairs varies widely, since the algorithm may sample a partner not trained for that layout; learning opportunities for Forced Coordination are particularly sparse, since Forced Coordination requires the partner's contribution to make any deliveries. Impressively, RACCOON can succeed when learning opportunities are sparse, where other methods fail—showing promise for RACCOON as a method to deal with more complex partner and task spaces, where identifying good learning opportunities by hand is infeasible.

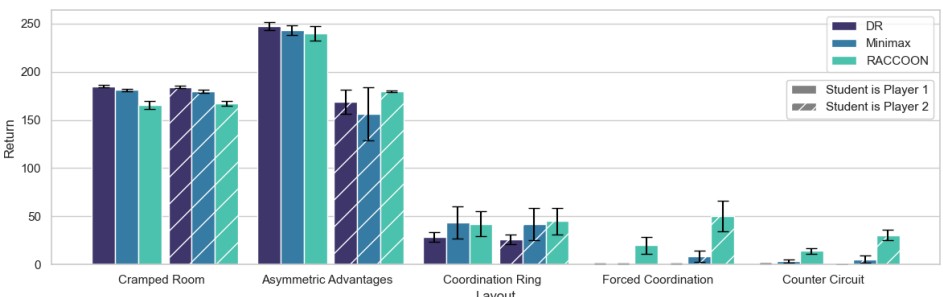

Figure 2: **Returns with held-out partners for students trained on all five layouts simultaneously.** Bars for each task show average returns with held-out partners trained to low, medium and high skill on that task. Students are trained for 400M environment interactions and the curriculum samples a partner, one of the five layouts and player assignments for each episode. Mean and standard error shown for five seeds.

Furthermore, the training curves shown in Figure 3 demonstrate that RACCOON is able both to converge quickly on the easier layouts (leftmost two), and prioritise learning on the remaining harder layouts.

**Analysis of curricula.** We provide an insight into the curriculum over *partners* induced by RAC-COON by considering the skill levels of partners sampled throughout training. For the multi-task setting, Figure 4 shows the probability of sampling training partners of low, medium and high skill levels for the relative regret and minimax scores. RACCOON initially prioritises high-skilled partners (who intuitively should be easier to cooperate with), and gradually increases the proportion of low-skilled partners throughout training. This is in line with preliminary findings by Bhati et al. (2023) that a curriculum *decreasing* in partner skill—i.e., in which a student initially trains with more skilled partners, followed by less skilled partners—is effective for the student's learning. In

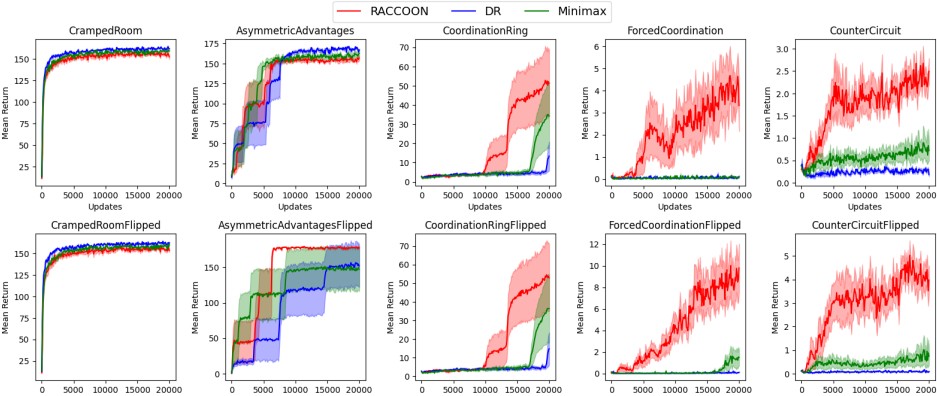

Figure 3: **Training returns on each task for students trained on all five layouts and both player positions.** Returns on each task are averaged over all training partners (which is why the returns for Forced Coordination and Counter Circuit appear low). RACCOON demonstrates faster convergence on easier tasks, and continues to learn throughout the curriculum. Mean and standard error shown for five seeds.

comparison, the Minimax curriculum maintains a much closer to even split (marked by the dotted grey line), suggesting it struggles to find efficient learning opportunities.

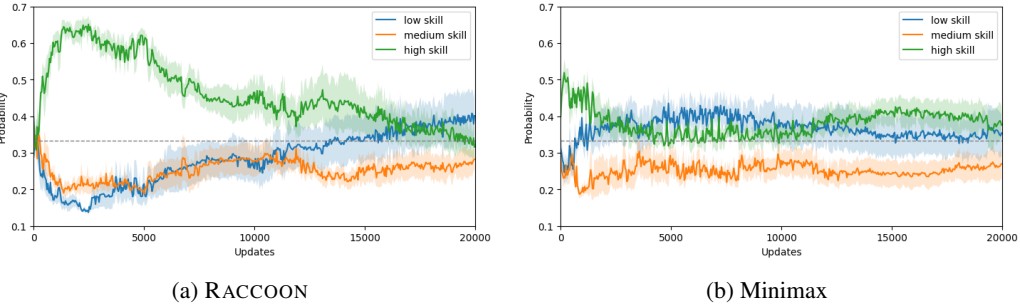

        (a) RACCOON                                 (b) Minimax

Figure 4: **RACCOON induces a curriculum which prioritises high-skilled, then low-skilled, partners.** Curves show the probability of sampling a partner of each skill level (for any task) throughout training, for a student training on all tasks.

For an insight into how RACCOON prioritises tasks, Figure 5 shows the proportions of each task sampled throughout training. We see that RACCOON is quick to identify Forced Coordination—the hardest task—as a layout to prioritise, while downweighting the easier layouts Cramped Room and Asymmetric Advantages. In addition, while the proportion of Counter Circuit is low, the training curves in Figure 3 show that the RACCOON student is still improving on Counter Circuit, suggesting that RACCOON is finding efficient task-partner pairs for learning on Counter Circuit, even when not sampling the layout as much as others. Appendix B.1 shows more analysis of individual partners sampled.

**Relative vs. absolute regret.** We hypothesised that using a normalised regret score in RACCOON is important for achieving robust performance across tasks and partners which may vary widely in their maximum achievable returns. To test this, we run an ablation which uses *absolute* rather than relative regret, which we denote by RACCOON⁻. The final performances of all four methods in the multi-task setting, including RACCOON⁻, are shown in Table 1 for comparison. We find that RACCOON⁻, like DR and Minimax, is unable to learn to cooperate on the hardest tasks. It is interesting to note that RACCOON achieves the highest return on all layouts except the easiest, Cramped Room—suggesting that there is some trade-off in performance across tasks. DR tends to overfit to the easiest tasks, while

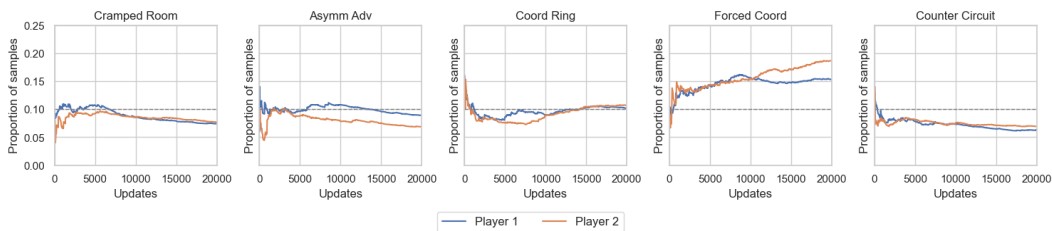

Figure 5: **RACCOON prioritises the hardest task.** Curves show the proportion of tasks of each type sampled throughout training, for one run of RACCOON training on all tasks (smoothed). RACCOON increases the probability of sampling Forced Coordination, the hardest task, throughout the run.

RACCOON with relative regret ensures the student learns to cooperate reasonably well with everyone, even if it doesn't achieve the highest performance in every case—arguably a preferable outcome.

**Scalable sample efficiency.** In addition to performance gains, we test the sample efficiency of RACCOON compared with DR by varying the number of training partners for a single task. We train students for 200M environment steps on Counter Circuit (Player 1) with 15, 30, 45 and 60 training partners (respectively 5, 10, 15 and 20 seeds and their checkpoints).

The training curves in Figure 6 show that RACCOON almost perfectly maintains sample efficiency as we scale the number of training partners, while the sample efficiency of DR rapidly degrades as number of partners increases. This implies that RACCOON is efficiently able to filter a large partner population to find the best learning opportunities, which not only makes it more effective in terms of compute, but makes it a promising method for dealing with large and potentially noisy partner populations in more complex settings where we may have less control over the quality of partners.

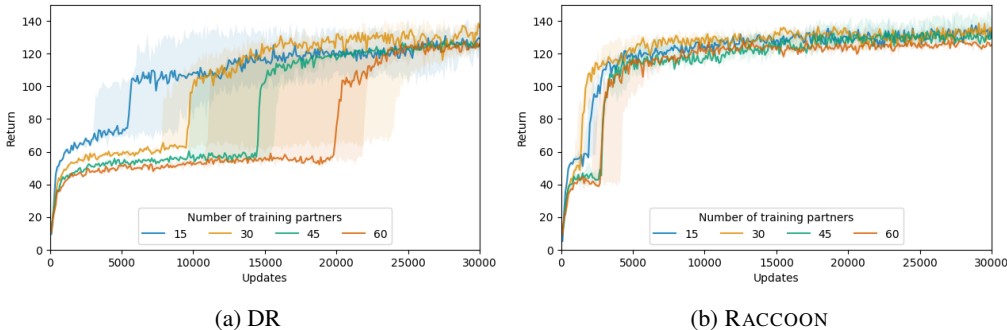

(a) DR    (b) RACCOON

Figure 6: **RACCOON maintains performance when scaling number of training partners.** Curves show training returns, averaged across training partners, for DR and RACCOON training on Counter Circuit only. As number of partners increases, DR requires many more environment interactions to achieve the same performance. Median and interquartile range shown for five seeds.

## 6 DISCUSSION

To our knowledge, our work is the first detailed investigation of autocurricula over partners for ad-hoc teamwork in cooperative MARL. Our experiments which train students on all layouts simultaneously show that RACCOON is able to navigate settings with diverse challenges and sparse learning opportunities—thereby learning a more versatile policy—where baselines fail to learn to solve harder tasks. In particular, we show that using *relative regret* is key for adapting to high variance in the difficulty of tasks. We additionally show that RACCOON maintains sample efficiency as we scale the number of partners, while that of DR steadily declines. While Overcooked is a relatively simple environment, these results in combination suggest that the benefits conferred by RACCOON may truly shine in large and complex task and partner spaces, where randomisation can have no hope of efficiently filtering useful learning opportunities.

**Limitations of the environment**   We test our method in the Overcooked environment because it presents distinct collaborative challenges while being relatively interpretable and fast to train in, and because it is commonly used in work on generating partner diversity for ad-hoc teamwork (Strouse et al., 2021; Sarkar et al., 2024; Zhao et al., 2023; Charakorn et al., 2022). However, the tasks in Overcooked are still much simpler than real-world collaborative tasks. Another limitation of Overcooked is that collaboration is not strictly necessary for reasonable performance on most of the five classic layouts, as evidenced by the high return achievable with random policies. This brings into question whether agents are truly learning meaningful collaborative skills, and whether the demonstrated efficacy of including low-skilled partners in the training population (Strouse et al., 2021) is due to improved cooperation or simply improved individual skill at the task.

**Limitations of cooperative regret**   In regret-based UED, a minimax regret policy will have regret at most equal to the minimax regret bound on all tasks in the training domain, but there are no guarantees on the behaviour of the policy on tasks where the regret is not at this bound. *Regret stagnation* occurs when it is possible to improve performance on levels which are not at the minimax regret bound, but the student no longer trains on those levels because the UED teacher only plays the highest regret levels (Beukman et al., 2024). This may be particularly problematic in cooperative settings, where diverse partners may employ *incompatible* policies such that, under uncertainty about its partner, the student is unable to simultaneously achieve low regret with all partners. While Overcooked is a relatively simple environment and good ad-hoc teamwork seems feasible, the problem of incompatible policies is especially notable in Hanabi (Cui et al., 2022), and we encourage further work to investigate and mitigate the limitations of regret in such settings.

**Limitations of relative regret score**   One limitation of the score described in Section 3.3 is that using the maximum return *ever* achieved with a partner on a task as an estimate for optimal return doesn't account for noise in the rewards. For example, it is occasionally possible to make a single delivery with a randomly behaving partner in Forced Coordination if the partner happens to take the right sequence of actions, even though most of the time it's impossible to collaborate with such a partner. As a result, relative regret would score such a partner highly, even though they don't provide meaningful learning. A more nuanced regret estimate should identify such situations and correctly prioritise only partners with whom cooperation is meaningfully possible.

**Future work**   A natural follow-up is to pair RACCOON with different partner population generation methods and larger environment spaces, particularly open-ended or procedurally generated environments (Fontaine et al., 2021). It would also be interesting to test the robustness of RACCOON with adversarial partner populations, for example those which are particularly sparse on useful learning opportunities or may contain "rogue" partners. Finally, a vision for future work is to *combine* the processes of training the partners and the student into a single adaptive process, for example by using learning potential to inform partner generation and selection in an evolutionary search method such as MAP-Elites (Mouret & Clune, 2015; Xue et al., 2022; Parker-Holder et al., 2020).

**Impact statement**   Our work aims to make autonomous agents better at collaborating with and adapting to partners on problems where all agents have a common goal, which can make them more helpful to humans, both in collaborating with humans directly or more efficiently achieving goals specified by humans for an autonomous multi-agent system. It's worth noting that, in general, agents trained purely to be cooperative may be vulnerable to exploitation when transferred to real-world settings, but we believe that at this stage—given the simplicity of the simulations in which we are working—our work does not pose any apparent negative consequences.

## 7   RELATED WORK

**Diverse partners for ad-hoc teamwork**   A number of methods have been used to generate diverse training partners for cooperation. A simple but effective method is Fictitious Co-Play (Strouse et al., 2021), which forms a population from different seeds and checkpoints of agents trained in SP. This is the method we use to train the partners for our experiments. Other methods use an auxiliary loss to regularise diversity. TrajeDi (Lupu et al., 2021) trains agents in SP while maximising a statistical divergence term between trajectories, while LIPO (Charakorn et al., 2022), CoMeDi (Sarkar et al., 2024), ADVERSITY (Cui et al., 2022) and BRDiv (Rahman et al., 2022) model policy compatibility

as XP return and aim to minimise this. However, unlike RACCOON, these methods train a best response student by sampling partners from the population uniformly at random. Notably, RACCOON is not intended as a competitor to these methods: indeed, it can be paired with any of these partner populations, and therefore *complements* work on diverse partner generation.

Maximum Entropy Population-Based Training (Zhao et al., 2023) uses a population entropy bonus to promote diversity, and uses a prioritised sampling method for sampling partners during the training of the student. However, unlike our method, partners with *lowest XP return* are prioritised, corresponding to a minimax adversary, which we show is less robust than regret as it does not reflect whether cooperation is achievable with each partner, in particular performing poorly on Forced Coordination.

**Curricula in cooperative MARL**    Automatic curriculum learning (Leibo et al., 2019) has enjoyed successes across RL. Self-play—in which an agent plays against itself, typically in a competitive setting (Tesauro et al., 1995)—forms a natural curriculum of increasing difficulty as the agent improves, and has been effective in achieving human-level performance in a number of games (Silver et al., 2017; Bakhtin et al., 2022). Fictitious self-play (FSP, Heinrich et al. (2015)) uses a uniform mixture of past checkpoints as training partners to avoid cycles and forgetting. Vinyals et al. (2019) introduced *prioritised* fictitious self-play (PFSP), which weights FSP partners as a function of the probability of beating them to identify partners which provide the best learning signal. While the motivation behind PFSP is similar to RACCOON, PFSP is applied in a competitive setting, whereas RACCOON is designed for cooperative settings and uses pre-trained rather than FSP partners. Similarly, MAESTRO (Samvelyan et al., 2023) combines UED and self-play to induce a curriculum over joint partner-task pairs, and is the most similar to RACCOON, again with the major difference that we use pre-trained partners for collaborative tasks. Additionally, MAESTRO scores partners and tasks using an estimate of absolute regret, rather than relative regret, as we do.

Bhati et al. (2023) also investigate teammate selection in Overcooked, but use hand-crafted curricula over skill levels in a single easy layout, whereas our curriculum automatically adapts to the student throughout training and is compatible with multiple tasks.

## 8    CONCLUSION

In this work, we introduce RACCOON, a novel cooperative autocurriculum method which provides a complement to diverse partner generation for ad-hoc teamwork. In particular, we define a *relative regret* metric to score the learning potential of partners and tasks. We demonstrate the increased robustness of RACCOON in the Overcooked environment by presenting diverse partners and tasks, finding that RACCOON learns to collaborate on the most difficult tasks where baselines fail. Our results on sample efficiency further suggest that RACCOON remains effective as we scale partner populations and task spaces. We conclude by discussing limitations of regret as well as potential avenues for future work. We hope that our work demonstrates the unexplored potential of autocurricula in cooperative MARL, and provides initial methods and results on which future work can build.

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

## A   ALTERNATIVE METHODS FOR ESTIMATING COOPERATIVE REGRET

In addition to the relative regret metric outlined in 3.3, we considered and tested a number of methods for estimating cooperative regret with a partner.

**Training a best response policy**   Since we want to estimate optimal return possible with each partner, we could try to train a best response policy $BR(\pi_i)$ to $\pi_i$ for each $\pi_i \in \Pi^{\text{train}}$, and use the maximum return achieved as the estimate of optimal return with $\pi_i$. While this was fairly effective for easier layouts, we found that this approach frequently underestimates optimal return with each partner, leading to negative scores—the student trained with $\Pi^{\text{train}}$ generally outperforms the best response policy with each training partner $\pi_i$ despite (or rather, because of) being trained with all partners.

**Treating the partner's SP as a best response**   Another approximation is to treat a partner's SP return as the optimal return with that partner. While this is more likely to be accurate when training with highly skilled (i.e., converged partners), it falls short in cases where partners are low-skilled (i.e., partially trained), on tasks where it is possible for one of the team members to "pick up the slack" of a less skilled partner. As a result, we found this method to be effective only on Forced Coordination, which does require both partners' contributions.

## B   ADDITIONAL EXPERIMENTAL RESULTS

### B.1   FURTHER CURRICULUM ANALYSIS

To give an example of which particular partners are most sampled by RACCOON, we provide Figure 7, which shows the cumulative samples of partners of each type for one run of RACCOON on all layouts with the ten most sampled partners labelled. We see that a large proportion of the partners sampled are medium and high-skilled Forced Coordination partners, which indicates that RACCOON is correctly identifying these as useful partners to learn from—since Forced Coordination requires both partners to work together to deliver a meal, achieving non-zero returns requires sampling a partner who has been trained on Forced Coordination.

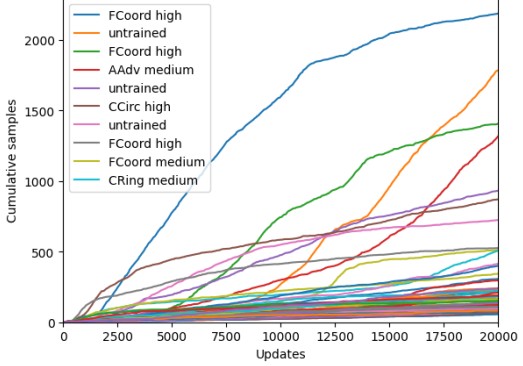

Figure 7: **Cumulative samples of individual partners by RACCOON training on all tasks.** Results from one run. Legend shows ten most sampled partners, identified by the layout they were trained on and their skill level (untrained refers to a low-skilled partner, i.e., random policy). RACCOON prioritises a high proportion of Forced Coordination partners, which correspond to the most challenging task.

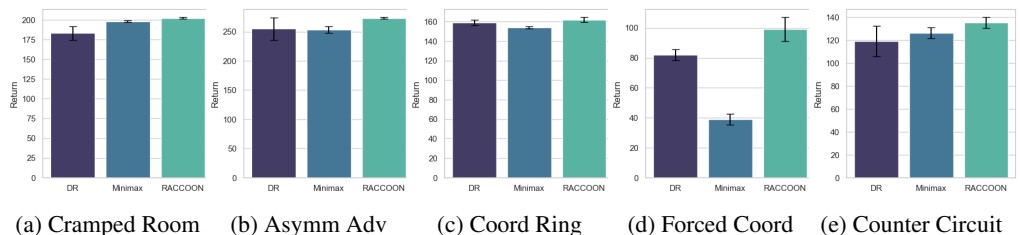

(a) Cramped Room    (b) Asymm Adv    (c) Coord Ring    (d) Forced Coord    (e) Counter Circuit

Figure 8: **Returns with held-out partners for single-task settings.** Students are trained on individual layouts with 15 training partners for 32M environment steps, and RACCOON achieves the highest return in all cases. Mean and standard error for five seeds shown.

## B.2 SINGLE TASKS

We provide additional results from training the student on single tasks (single layout with student as Player 1). For each experiment on layout $L$, we use 15 training partners consisting of three checkpoints of five seeds from the pre-trained specialist policies $\Pi_L^{\text{train}}$, and train the student for 32M environment steps. Figure 8 shows test performance held-out partners $\Pi_L^{\text{test}}$ for each student trained on $L$. RACCOON modestly outperforms DR and Minimax on easier tasks (a)-(c) and substantially outperforms both on the hardest layout (d)—the only layout in which a player working alone cannot deliver soup. As predicted, Minimax performs particularly poorly on (d), as it is impossible to succeed with low-skilled partners except through chance.

However, we note that the advantage conferred by RACCOON in this setting is marginal, likely because the problem is sufficiently simple that DR and Minimax are already effective.

Figure 9 shows the overall percentage of each skill level sampled by RACCOON for each layout in the single-task setting, and we see that in most cases RACCOON prioritises low- and medium-skilled partners, intuitively because they are more difficult to collaborate with.

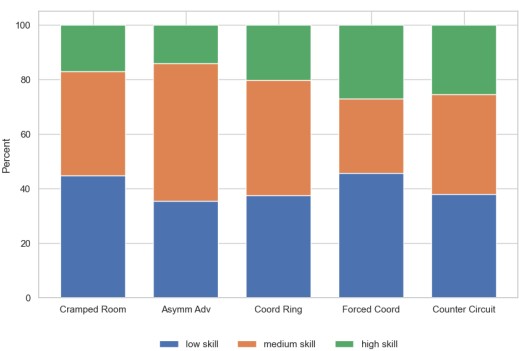

Figure 9: **Overall proportion of partners of each skill level sampled by RACCOON in single-task setting.** Mean shown over five seeds.

## C IMPLEMENTATION DETAILS

### C.1 ENVIRONMENT DETAILS

We use the JaxMARL implementation of Overcooked (Rutherford et al., 2024) as the base of our environment. We pad all layouts to shape $6 \times 9$, and agents receive an observation of shape $6 \times 9 \times 26$, where the 26 mostly binary channels represent positions and states of objects in the map, such as onion and pot locations, number of onions in a pot and remaining pot cooking time. Both players receive a reward of +20 when a soup delivery is made. In addition, for training SP partners for Coordination Ring, Counter Circuit and Forced Coordination, we shape the reward with an additional reward of +1 for placing an onion in a pot, which we anneal over 2.5M environment steps.

## C.2 AGENT ARCHITECTURE

Both partners and students use the same neural architecture, consisting of actor and critic networks with a shared convolutional backbone. The input observation is passed through a convolutional layer with 16 filters and kernel size 3. The processed observation is then passed to fully-connected policy and value heads, each with one hidden layer of dimension 32, to output action logits and values.

## C.3 HYPERPARAMETERS

### C.3.1 SELF-PLAY PARTNERS

Partners were trained via SP on individual layouts for 4.8M environment steps with checkpoints taken every 96,000 steps, and checkpoints for the training population were selected from the beginning of training, when performance converged, and the point at which return was half the final return. All partners were trained using 32 parallel environments, 300 rollout steps per update, discount factor 0.99, GAE-$\lambda$ 0.95, value loss 0.5, max grad norm 0.5, 4 minibatches per epoch, 5 PPO epochs and PPO clip epsilon 0.2. The hyperparameters which differed between layouts are shown in Table 2. We trained 50 seeds per layout but didn't use all of them in experiments.

Table 2: Hyperparameters for SP policies trained on each layout.

| Hyperparameter | Cramped Room | Asymm Adv | Coord Ring | Forced Coord | Counter Circuit |
|---|---|---|---|---|---|
| Learning rate | 0.0005 | 0.001 | 0.0005 | 0.002 | 0.002 |
| Entropy coefficient | 0.01 | 0.001 | 0.01 | 0.02 | 0.02 |

### C.3.2 STUDENTS

Hyperparameters for students trained on all layouts are shown in Table 3.

Table 3: Student hyperparameters for training on all layouts (Section 5)

| Hyperparameter | Value |
|---|---|
| Max episode steps | 400 |
| $\gamma$ | 0.99 |
| $\lambda_{\text{GAE}}$ | 0.95 |
| Learning rate | 0.001 |
| Parallel environments | 50 |
| Entropy coefficient | 0.01 |
| Value loss coeffcient | 0.5 |
| Max grad norm | 0.5 |
| Total updates | 20,000 |
| PPO clip eps | 0.2 |
| PPO rollout length | 400 |
| PPO epochs | 4 |
| Minibatches per epoch | 4 |
| Buffer size | 50 |
| Replay probability | 0.9 |

For training on individual layouts, we used the hyperparameters in Table 4, and a partner sampling temperature of 1 for all layouts except Forced Coordination, for which we use temperature 3.

## C.4 COMPUTE

All experiments were conducted on a single NVIDIA RTX A6000 GPU with 48GB of VRAM, 500GB storage and 55GB RAM allocated with 6 vCPUs. A total of 1,000 hours of compute time were

Table 4: Student hyperparameters for training on individual layouts (Appendix B.2)

| Hyperparameter | Value |
|---|---|
| Max episode steps | 400 |
| $\gamma$ | 0.99 |
| $\lambda_{\mathrm{GAE}}$ | 0.95 |
| Learning rate | 0.001 |
| Parallel environments | 16 |
| Entropy coefficient | 0.01 |
| Value loss coeffcient | 0.5 |
| Max grad norm | 0.5 |
| Total updates | 5,000 |
| PPO clip eps | 0.2 |
| PPO rollout length | 400 |
| PPO epochs | 4 |
| Minibatches per epoch | 4 |
| Buffer size | 8 |
| Replay probability | 0.8 |

used for the project, including failed attempts and hyperparameter tuning, with the results presented in the paper taking fewer than 100 hours of that time.

For runs of RACCOON, DR and Minimax, 100M environment steps take 20-35 minutes to run, depending on the number of parallel workers.

