# OpenReview forum: "RACCOON: Regret-based Adaptive Curricula for Cooperation"
_ICLR.cc/2025/Conference — Submitted to ICLR 2025_

### Official Review · Reviewer_rKmk · 2024-11-02

**Soundness:** 3
**Presentation:** 2
**Contribution:** 2
**Rating:** 6
**Confidence:** 3

**Summary:**

This paper argues that partner sampling is an important yet overlooked issue when training agents to cooperate with novel partners. To address this, the authors propose Regret-based Adaptive Curricula for Cooperation (RACCOON), which prioritizes high-regret partners and tasks. This approach allows adaptation to the student (the learning agent)'s changing abilities throughout training and reflects the relative learning potential of each (partner, task) pair for the student. RACCOON demonstrates improvements on challenging tasks within the Overcooked environment. The paper analyzes the method through experiments involving varying skill levels, task difficulties, and scalability.

**Strengths:**

1. The issue of partner sampling is crucial for the generalizability of MARL and broader multi-agent collaboration algorithms.
2. The paper compares the proposed method with other mainstream sampling approaches and extends the experiments with deeper analysis.
3. While the concept of regret is not novel in reinforcement learning, its application to partner sampling is a new contribution.

**Weaknesses:**

1. The paper adopts "the difference between the maximum return ever achieved with $\pi'$ (on that task) and the current return" as the regret estimation. I believe this approach is relatively simplistic, and the performance of this method may depend on an effective exploration strategy, which is not adequately addressed in the paper. The authors might consider exploring better estimation methods and comparing their effectiveness.
2. The readability of the paper could be improved, as some figures are of low quality and the text is difficult to follow.
3. The experiments involve only two agents; cooperation with multiple novel partners would present greater challenges, and the proposed method may not be directly applicable in such scenarios. The paper does not seem to address this or discuss the limitation of this aspect.

**Questions:**

1. Why does the return of Asymm Adv in Figure 3 suddenly increase?
2. Why are the returns for Forced Coordination and Counter Circuit in Figure 3 measured in different ways?
3. How are the skill levels defined?
4. Why choose the very challenging task Counter Circuit to conduct scalable sample efficiency experiments in Figure 6?
5. How can the proposed methods be scaled to tasks involving cooperation with two or more partners?

---

> ### Author Response · Authors · 2024-11-18
> **Addressing weaknesses and answering questions**
>
> We thank the reviewer for their insightful feedback. We address the raised weaknesses in turn below.
>
> **Regret estimate:** We use the difference between the maximum return ever achieved on a (partner, task) pair and the current return as our regret estimate. We do also discuss *alternative methods for estimating regret* in Appendix A. The simplicity of our metric is one of its strengths; if regret on (partner, task) is defined as A-B, where A is the maximum achievable expected return on (partner, task), then estimating A more directly would require, for example, training a single best response to each (partner, task) to estimate the maximum achievable returns with that (partner, task). For our approach, we don’t need to incur such costs, instead only needing to store metrics we already have from the run so far. In addition, we posit that there may be an advantage to using the maximum return achieved so far, rather than the maximum return ever achievable, since this scales to the student’s current ability.
>
> **Low quality figures:** We thank the reviewer for pointing this out. We find this somewhat surprising, as reviews on the whole seem to have found the paper clear and well-presented. We ask which of the figures, and which parts of the text, the reviewer found unclear, so that we can directly provide clarifications and strengthen the written communication of the paper.
>
> **Scaling to more agents:** We acknowledge that the paper is currently limited to the two-player setting, which has been studied throughout much of the ad-hoc coordination literature [1, 2, 3]. The two-agent setting has different, but not necessarily less interesting dynamics than the > 2-player setting, as when there is only one partner with which the agent can coordinate, they have more potential to shape the behaviour of the other agent and influence equilibrium selection. With greater numbers of agents, the coordination problem for an individual agent increasingly reduces to "fitting in" with the group. We have added this point to the paragraph on “Environment” in Section 4 of the manuscript (coloured in red). This being said, there is certainly scope for RACCOON to be scaled to more partners. How this is done depends on the setting and whether partners are interchangeable; one option which is less computationally costly than simply increasing the number of dimensions of the buffer is to use pre-trained teams of partners, and use RACCOON to sample teams and tasks.
>
> ### Answering Questions:
>
> 1. The jumps in training returns in Figure 3 likely arise from the student acquiring new skills and learning to cooperate with new partners.
>
> 2. We apologise for unclear wording; all the plots in Figure 3 show the training return averaged across all partners, and we merely intend to point out that this is the reason for the returns for Forced Coordination and Counter Circuit being so low (since the student gets 0 return with most partners on these tasks). We have amended the wording in the manuscript to improve clarity.
>
> 3. Section 4 describes how partners are generated. Partners of different skill levels are obtained using checkpoints of policies from the beginning, middle and end of training, following prior work [1]. More specifically, partners are trained in self-play until their returns converge. These final policies are the “high-skilled” partners. The “medium-skilled” partners use the checkpoint which achieves half of the final return. The “low-skilled” partners use the checkpoint at initialisation.
>
> 4. We choose to use Counter Circuit to analyse effects of scaling the number of training partners because it is challenging enough that we expect partners to exhibit more diverse policies than for an easier layout such as Cramped Room. In particular, training a student on a single, easy layout such as Cramped Room is almost trivial, and the task is too easy to present additional challenges as the student has to deal with additional training partners.
>
> 5. See “scaling to more agents” above.
>
> [1] Strouse, D. J., et al. "Collaborating with humans without human data." Advances in Neural Information Processing Systems 34 (2021): 14502-14515.
>
> [2] Li, Yang, et al. "Cooperative Open-ended Learning Framework for Zero-Shot Coordination." The 40th International Conference on Machine Learning. 2023.
>
> [3] Wang, Rose, et al. "Too many cooks: Coordinating multi-agent collaboration through inverse planning." The 9th International Conference on Autonomous Agents and Multi-Agent Systems. 2020.

---

> > ### Comment · Reviewer_rKmk · 2024-11-25
> >
> > I appreciate the detailed response. Most of my concerns are resolved. The figure quality has been fixed in the revision. I will raise the score to 6.

---

### Official Review · Reviewer_8V6B · 2024-11-02

**Soundness:** 3
**Presentation:** 4
**Contribution:** 4
**Rating:** 8
**Confidence:** 4

**Summary:**

Prior work on UED has used the idea of "regret-minimization" as a promising way to generate a curriculum of tasks for agents to train on. For building agents capable of zero-shot coordination, there is a lot of MARL research on generating diverse partners to train with. However, there is not much work on which partners to train with. This paper introduces a regret-based curriculum for deciding which partners in a population are most useful for training with, and compares its approach to common objectives from the UED literature.

**Strengths:**

This is a pretty original idea which has promising applications. The idea to create a curriculum for training against population based agents is very interesting and the authors did a good job showing the sample efficiency and learning trends of agents using their RACCOON approach. I particularly thought the analysis of sample probability as a function of updates was compelling, both for partner skill level and problems sampled, was compelling and revealed some of the limitations of naively choosing which partner from a population to train with. Moreover, the algorithm was very cleanly written and easy to follow, and the tensions between relative vs absolute regret, and other methods for formalizing regret in the appendix, was well motivated and ablated well.

**Weaknesses:**

It was a little unclear at first that the evaluations for the main section of the paper in section 5 were done on a multi-task setting, particularly when prior overcooked work has looked at just single-task performance. While the single task problems were still introduced, to someone quickly looking at results like Figure 2, it may be confusing that traditional PBT methods (essentially Domain Randomization for just partners from my understanding) did not replicate in grids like Forced coordination. These methods do replicate in the results shown in the appendix, so maybe reference this earlier. I would emphasize in the "Tasks" section of Section 4 that you will primarily be addressing the multi-task setting, and for the caption of Figure 2 add a brief line saying that by "all tasks" agents are trained on you mean they could sample any of the 5 grids in addition to partners. If you have space, I would briefly describe how your method stacks up to DR and Minimax on the single task section right at the beginning of section 5 so that someone reading it quickly can understand how your approach can still replicate results from the canonical study of Overcooked.

Moreover, for analysis as to why the multi-task section is failing for harder grids, the plots of grids sampled is nice, but even more interesting would be a look at the state the agents are failing at. Is it just because agents trained with DR are learning a different set of norms compared to RACCOON that don't support generalization, or is it something much simpler like they just don't know how to pass an item at the top of a grid compared to the bottom? If it's the latter, an explanation as to how RACCOON helps address this issue would be nice. If you could include some examples of states or sequences of states to compare the different failure modes of RACCOON and DR on harder grids like Forced Coordination that would make for a really strong qualitative analysis.

**Questions:**

Why are agents struggling in the multi-task setting but doing well in the single-task setting on the same grids, especially since your results show the harder grids are being sampled more for RACCOON?

---

> ### Author Response · Authors · 2024-11-18
> **Taking on feedback**
>
> We thank the reviewer for their feedback and are pleased they found our idea interesting and original, and were compelled by our results. We also appreciate the reviewer’s feedback on the clarity of our presentation - we are pleased that the algorithm was easy to follow, and it is useful to know that the switch between multi-task and single-task settings can be confusing at first glance. We have taken on the reviewer’s feedback on the presentation of our results in Section 5, and the corresponding modifications can be found in the manuscript (coloured in red). We address additional feedback below.
>
> **Showing states where agents fail:** We agree that it would be insightful to include qualitative analysis of agent failures by including visualisations of the states in which the behaviours of DR and RACCOON agents fail - however, at present we do not have the means to easily obtain these visualisations in time for the discussion period. We will endeavour to look into doing so for a camera-ready copy, but ask if there are any intermediate results that might provide qualitative insights of interest to the reviewer.
>
> **Why agents do better in the single-task setting for the same layout.** We attribute this to there being a trade-off in performance across all tasks for a fixed training budget. Since each task requires different skills to complete, it is more challenging for an agent to perform well in all tasks than just one of them. Therefore, the returns for an agent trained on a single task tend to be higher than returns on the same task when it’s one of multiple distinct tasks being trained on.

---

> > ### Comment · Reviewer_8V6B · 2024-11-24
> >
> > Thanks for offering some insight. If you could get those state visualizations in time for the camera ready that would be great, but otherwise very interesting idea!

---

### Official Review · Reviewer_ABvg · 2024-11-03

**Soundness:** 2
**Presentation:** 2
**Contribution:** 2
**Rating:** 5
**Confidence:** 4

**Summary:**

This paper proposes a new method for configuring curriculum tasks in multi-agent reinforcement learning (MARL) based on regrets. The primary aim of this approach is to address overfitting issues in MARL algorithms, which typically lack zero-shot generalization. In scenarios where partner behaviors change even slightly, existing algorithms perform significantly worse. The regret-based curriculum learning framework dynamically selects agents, encouraging the MARL algorithm to avoid memorizing environment-specific information solely for performance gains. Instead, it promotes adaptive strategies that utilize cooperative partners more effectively.

To evaluate their method, the authors conducted experiments in a collaborative MARL environment called OverCooked, focusing on its generalization capabilities. The results demonstrate that this curriculum-based approach facilitates a more generalizable and efficient method for leveraging partner information across diverse environments, yielding strong performance on the Forced Coord and Counter Circ tasks. Analysis of the training dynamics reveals that the proposed method initially prioritizes high skilled partners early in training, gradually shifting to lower-level partners toward the end, optimizing the agents’ adaptability.

**Strengths:**

- This paper addresses the overfitting challenges inherent in existing MARL approaches. Typically, standard MARL environments guide algorithms to train agents to perform specific actions in particular scenarios, often tailored to the environment itself, rather than utilizing cooperative behavior among agents. In contrast, this study focuses on improving inter-agent cooperation, proposing a method that promotes zero-shot generalization in MARL. This contribution is especially significant as it introduces a pathway for MARL algorithms to achieve more adaptable and generalized learning that extends beyond environment-specific contexts.

**Weaknesses:**

- The experimental design of this paper does not align well with the principles of MARL. Although the proposed method is at an early research stage, it is difficult to claim effectiveness in a multi-agent context based solely on a two-agent environment. To demonstrate robust, generalized performance, results should ideally involve three or more agents, which would provide a more comprehensive test of the method’s effectiveness. Limiting experiments on only two agents places significant constraints on the study’s findings.

- In a fully observable environment with two agents, the necessity of agent-based learning itself is questionable. In this setting, the benefits of using MARL over conventional reinforcement learning are unclear, as there is no substantial increase in state or observation complexity. This raises concerns about the rationale for applying an agent-based approach in these circumstances.

- The experimental explanation is also insufficient. While the OverCooked environment is used, the paper should clearly explain why this environment is particularly suitable for examining generalization, overfitting, and zero-shot evaluation. Providing this context would better support the experimental design and underscore its relevance to the paper.

**Questions:**

- The concept of calculating regret in RL raises questions about its alignment with the field’s core principles. RL traditionally focuses on learning directly from the environment via online learning techniques, so introducing a regret variable and selecting the best task based on this metric may diverge from standard RL practices. Although this paper addresses this issue by using pre-trained agent policies and concentrating solely on selecting effective tasks, this approach still appears somewhat misaligned with the philosophy of RL, which emphasizes learning through environmental interactions rather than pre-determined information.

- I would like to ask the proposed method could improve generalization performance if incorporating partially observable or noisy environments, rather than a fully observable setting. I believe that such stochasticity might actually help mitigate overfitting. Furthermore, insights into how performance might vary under these settings would add valuable perspective on the robustness and adaptability in diverse environments.

**Details Of Ethics Concerns:**

No concern

---

> ### Author Response · Authors · 2024-11-18
> **Addressing weaknesses and answering questions**
>
> We thank the reviewer for their detailed feedback, and address the weaknesses raised below.
>
> **Two-agent environment:** We acknowledge that the paper is currently limited to the two-player setting, which is studied throughout much of the ad-hoc teamwork literature through environments like Overcooked and Hanabi [1, 3, 4, 5, 6]. The two-agent setting has different, but not necessarily less interesting, dynamics than the setting with > 2 players, as when there is only one partner with whom the agent can coordinate, they have more potential to shape the behaviour of the other agent and influence equilibrium selection. With greater numbers of agents, the coordination problem for an individual agent increasingly reduces to "fitting in" with the group. However, RACCOON could be scaled to multiple partners by, for example, adding extra dimensions to the buffer, or by using teams of partners in place of individual partners in the algorithm.
>
> **Complexity of Overcooked:** Overcooked has been used as a benchmark throughout much of the ad-hoc teamwork literature precisely because it isolates the effective partial observability induced by not knowing a partner's policy or higher-level convention [1]. The coordination biases of a previously unseen partner are unobserved and can only be discovered through an interaction history. Introducing environment partial observability, as in the Hanabi challenge, does introduce further challenges worthy of research, however we do not focus on these in this paper.
>
> **Motivation for Overcooked:** We thank the reviewer for pointing out that we should include an explicit motivation for the experimental setting in the paper to improve clarity. We have added this to the “Environment” paragraph in Section 4 (coloured red). We appreciate that this will strengthen the paper and that it ties in with the other concerns raised.
>
> ### Answering Questions:
>
> 1. **Use of regret in RL:** Agents are still learning through environment interactions, and the statistics used to estimate regret are derived entirely from these interactions. As noted in [7, 8], curricula in reinforcement learning are intended to prioritise future experiences based on regularly updated regret statistics in order to improve sample efficiency. This has been shown to be immensely valuable in a field that consistently suffers from sample inefficiency.
>
> 2. **Extension to partial observability.** We acknowledge that the partially observable setting comes with its own host of challenges, making it a direction worthy of a distinct investigation. This is because of the potential for “irreducible regret”, which can arise when an agent cannot simultaneously be optimal with two distinct partners under partial observability, because they rely on different conventions [2]. This is beyond the scope of our paper but we acknowledge it as a fruitful direction for future research.
>
> [1] Sarkar, Bidipta, Andy Shih, and Dorsa Sadigh. "Diverse conventions for human-AI collaboration." Advances in Neural Information Processing Systems 36 (2024).
>
> [2] Beukman, Michael, et al. "Refining Minimax Regret for Unsupervised Environment Design." Forty-first International Conference on Machine Learning.
>
> [3] Strouse, D. J., et al. "Collaborating with humans without human data." Advances in Neural Information Processing Systems 34 (2021): 14502-14515.
>
> [4] Zhao, Rui, et al. "Maximum entropy population-based training for zero-shot human-ai coordination." Proceedings of the AAAI Conference on Artificial Intelligence. Vol. 37. No. 5. 2023.
>
> [5] Charakorn, Rujikorn, Poramate Manoonpong, and Nat Dilokthanakul. "Generating diverse cooperative agents by learning incompatible policies." The Eleventh International Conference on Learning Representations. 2023.
>
> [6] Cui, Brandon, et al. "Adversarial diversity in hanabi." The Eleventh International Conference on Learning Representations. 2023.
>
> [7] Jiang, Minqi, Edward Grefenstette, Tim Rocktäschel. "Prioritized level replay." The Thirty-Eighth International Conference on Machine Learning. 2021.
>
> [8] Dennis, Michael, et al. "Emergenet complexity and zero-shot transfer via unsupervised environment design." The Thirty-Fourth Conference on Neural Information Processing Systems. 2020.

---

> > ### Comment · Reviewer_ABvg · 2024-11-21
> > **Response to authors rebuttal**
> >
> > I appreciate the authors’ efforts to address my concerns. However, the current rebuttal does not fully resolve my issue regarding the simplicity of environments. While the proposed method demonstrates better performance in some cases, I still feel that the authors have not adequately shown that the method generalizes effectively to curriculum learning across multiple environments, especially in scenarios involving more than two agents.
> >
> > Given this, I maintain my current score.

---

### Official Review · Reviewer_nToe · 2024-11-03

**Soundness:** 4
**Presentation:** 4
**Contribution:** 2
**Rating:** 5
**Confidence:** 4

**Summary:**

The authors present RACCOON, a method for designing the curriculum for training generalist ad-hoc teamplay agents. The authors propose to sample training partners and tasks at each training iteration by ranking the partners based on relative regret. The authors apply RACCOON on a multi-layout Overcooked setting where a single student agent has to learn to generalize to new partners across multiple Overcooked layouts.

**Strengths:**

- The paper tackles a novel area in ZSC/Ad-hoc teamplay, autocurricula design.
- The proposed method is relatively straightforward and should be able to be easily plugged into most existing ZSC/Ad-hoc teamplay methods.
- The paper is generally well written and the authors present their motivation, methodology and results in a succinct manner.

**Weaknesses:**

- I am not convinced by the multi-task experimental set up where a single student agent is evaluated on all 5 Overcooked layout simultaneously. The papers results show that all baselines includeing RACCOON can barely deliver more than more than 2 dishes on the 3 more challenging layouts, suggesting that the agents did not learn any meaningful cooperative policies. Perhaps a more more interesting multi-task setup would be to move the locations of counters/pots around similar to what was proposed in [1].
- The authors use a very simple method of generating training partners (random initialization + adding poast checkpoints) without any explicit methods to encourage partner diversity when many such methods exist (TrajeDi, LIPO, CoMeDi etc.)


References:

[1] Ruhdorfer, C., Bortoletto, M., Penzkofer, A., & Bulling, A. (2024). The Overcooked Generalisation Challenge. arXiv preprint arXiv:2406.17949.

**Questions:**

- I would like to know authors' rationale for proposing to sample the tasks  _after_ sampling the partner. The authors mention that all partners are "specialist agents" and sampled partner might be used to generate an episode that they are not trained on. Wouldn't it more effective to first sample the task and then sample the partners that are trained on said task?
- As the student's policies improve, wouldn't it result in a negative score based on the equation presented in Section 3.3?

---

> ### Author Response · Authors · 2024-11-18
> **Addressing weaknesses and answering questions**
>
> We thank the reviewer for their feedback, and appreciate that they found the paper well-written. We address the weaknesses outlined below.
>
> **Multi-task experimental set up:** We use the five Overcooked layouts because they present a diverse range of challenges - in particular they are designed to provide different cooperative challenges (such as division of labour and avoiding collisions), so performing well on all of them is highly non-trivial. In addition, using these five fixed tasks makes it easier to obtain partners of particular, known skill levels on each task, making the performance of the algorithm more interpretable. However, beyond using this set-up to demonstrate the effectiveness of RACCOON, unsupervised environment design methods such as RACCOON are indeed particularly well-suited to procedurally generated environment spaces, so a fruitful step for follow-up work would indeed be to apply RACCOON with a more open-ended environment generator, such as [3].
>
> **Partner generation:** While a range of partner generation methods exist, fictitious co-play [1], the method we follow, has been shown to be effective for diverse partner generation despite its simplicity [2]. In addition, an advantage of using a regret-based autocurriculum is its ability to automatically discover the partners with whom it’s challenging to cooperate, which will implicitly reflect partner diversity  without us needing to know in advance which conventions each partner follows.
>
> ### Answering Questions:
>
> 1. **Sampling tasks after partners:** We design RACCOON to be applicable in the most general cases where we may not have access to privileged knowledge about which partners are best at which task (and it might be difficult to know this a priori); the power of RACCOON is that it can automatically discover which tasks are most useful to train on with each partner. Therefore, we didn’t restrict the algorithm to sample only (partner, task) pairs where the partner was explicitly trained on the task, instead allowing the algorithm to discover high-regret (partner, task) pairs for itself. The decision to sample a partner first and then a task given that partner, rather than vice versa, is because the size of the population of partners is fixed, whereas in procedurally generated environments there may be arbitrarily many tasks, and it would be infeasible to maintain a buffer for every possible task.
>
> 2. **Negative scores:** While it is possible for scores to dip below zero, doing so would immediately update the stored “maximum return achieved” and therefore restore the scores to non-negative in the next iteration.
>
> [1] Strouse, D. J., et al. "Collaborating with humans without human data." Advances in Neural Information Processing Systems 34 (2021): 14502-14515.
>
> [2] Charakorn, Rujikorn, Poramate Manoonpong, and Nat Dilokthanakul. "Investigating partner diversification methods in cooperative multi-agent deep reinforcement learning." Neural Information Processing: 27th International Conference, ICONIP 2020, Bangkok, Thailand, November 18–22, 2020, Proceedings, Part V 27. Springer International Publishing, 2020.
>
> [3] Ruhdorfer, Constantin, et al. "The Overcooked Generalisation Challenge." CoRR (2024).

---

> ### Comment · Reviewer_nToe · 2024-11-22
>
> I would like to thank the authors for replying to my questions, particularly on the motivation behind the design and experimental setup of the proposed method. However, I am still not convinced that the proposed method can effectively generalize to the multi-task setup, considering the low performance of the 3 harder layouts. Hence I will maintain my current score.

---

### Author Response · Authors · 2024-11-18
**Updates to the manuscript and individual comments**

We would like to thank all reviewers for their insights on the paper and suggestions for improvement. We have taken on all actionable suggestions and updated the manuscript accordingly. We have also submitted individual comments to each reviewer with further details relevant to their specific reviews.

We thank each reviewer in advance for taking the time to read our comments engage in further discussion.

---

### Meta-Review · Area_Chair_mEkD · 2024-12-23

**Metareview:**

This paper proposes a new method for configuring curriculum tasks in multi-agent reinforcement learning (MARL) based on regrets. The primary aim of this approach is to address overfitting issues in MARL algorithms, which typically lack zero-shot generalization. The regret-based curriculum learning framework dynamically selects agents, encouraging the MARL algorithm to avoid memorizing environment-specific information solely for performance gains. Instead, it promotes adaptive strategies that utilize cooperative partners more effectively. To evaluate their method, the authors conducted experiments in OverCooked. The results demonstrate that this curriculum-based approach facilitates a more generalizable and efficient method for leveraging partner information across diverse environments, yielding strong performance on the Forced Coord and Counter Circ tasks. Analysis of the training dynamics reveals that the proposed method initially prioritizes high skilled partners early in training, gradually shifting to lower-level partners toward the end, optimizing the agents’ adaptability.

All reviewers and the AC believe this paper studies an important problem, and the proposed approach is interesting. However, there were concerns about the experiments: (1) the multi-task setup made all baselines and RACCOON particularly weak, and (2) the environments only involved two agents. The AC agrees with these concerns and thus recommends rejection.

**Additional Comments On Reviewer Discussion:**

There were concerns about the experiments: (1) the multi-task setup made all baselines and RACCOON particularly weak, and (2) the environments only involved two agents. These concerns were not fully addressed in the rebuttal.

---

### Decision · Program_Chairs · 2025-01-22

Reject